# Support Vector Machines with Quantum State Discrimination

**Roberto Leporini** [1,*] and **Davide Pastorello** [2]

1  Department of Economics, University of Bergamo, Via dei Caniana 2, I-24127 Bergamo, Italy
2  Department of Information Engineering and Computer Science, University of Trento, Via Sommarive 9, I-38123 Povo, Italy; d.pastorello@unitn.it
*  Correspondence: roberto.leporini@unibg.it

**Abstract:** We analyze possible connections between quantum-inspired classifications and support vector machines. Quantum state discrimination and optimal quantum measurement are useful tools for classification problems. In order to use these tools, feature vectors have to be encoded in quantum states represented by density operators. Classification algorithms inspired by quantum state discrimination and implemented on classic computers have been recently proposed. We focus on the implementation of a known quantum-inspired classifier based on Helstrom state discrimination showing its connection with support vector machines and how to make the classification more efficient in terms of space and time acting on quantum encoding. In some cases, traditional methods provide better results. Moreover, we discuss the quantum-inspired nearest mean classification.

**Keywords:** quantum state discrimination; optimal quantum measurement; support vector machine; machine learning





## 1. Introduction

Support vector machines are becoming popular in a wide variety of applications [1]. They are supervised learning models with associated algorithms (such as sub-gradient descent and coordinate descent) that analyze data for classification [2]. A support vector machine (SVM, for short) learns by examples to assign labels to feature vectors. An object with a feature vector is treated through a kernel function as a point in a larger space and the goal is to find the maximum-margin separating hyperplanes that allow one to partition the space and divide the points into classes to which labels are assigned. The logic behind the kernel function of an SVM and the kernel methods in general turns out to be rather similar to what is seen in quantum computing when one performs an encoding of classical data into quantum states. In fact, quantum computing provides implicit computations in high-dimensional Hilbert spaces by means of the physical manipulation of quantum systems, as well as kernel methods provide implicit computation in a higher dimensional feature space by means of the efficient representation of inputs. The interpretation of quantum encodings as feature maps with relevance in quantum machine learning is well-established [3,4], and it is one of the crucial points of this work. In addition to the general connection between kernel methods and quantum computing, there are the expressive quantum approaches to SVM in terms of implementations of this model on quantum computers. For instance, in [5], the authors propose a discretized SVM whose training is performed applying the Grover algorithm. The celebrated proposal of quantum SVM by Rebentrost, Mohseni and Lloyd [6] is based on data retrieval from a quantum random access memory, the quantum phase estimation algorithm and the SWAP test. The resulting quantum algorithm allows a direct implementation of polynomial kernels in terms of tensor products of the quantum states encoding the training vectors. Furthermore, a quantum implementation of SVM on a quantum annealer has been recently proposed [7].

Quantum structures can be used to devise novel machine learning algorithms that do not require quantum hardware in the sense that the mathematical formalism of quantum

mechanics is applied to deal with data that are managed by classical computers. The so-called *quantum-inspired machine learning* is based on particular kinds of information storing and processing defined by means of objects from the quantum formalism that do not necessarily represent physical quantum systems. In the context of quantum-inspired machine learning, SVM has been studied [8], and the present work focuses on a quantum-inspired classification algorithm that turns out to be similar to an SVM.

The general idea of classification algorithms based on discrimination of quantum states is also supported by recent experimental works on quantum state classification based on classical machine learning methods, such as the proposals in [9–13], for instance. In [14], the authors demonstrate a machine learning approach to construct a classifier of quantum states training a neural network. In [15], convolutional neural networks and principal component analysis are applied to classify polarization patterns in quantum optics.

An interesting quantum-inspired binary classification algorithm has been introduced in [16] in terms of a nearest mean classifier based on trace distance between density operators encoding feature vectors. To handle multi-class with this binary classifier, there are different techniques: *one against one*, which constructs a classifier for each pair of classes, *one against all*, which builds one per class, *hierarchical classification*, which creates a tree, where the leaves correspond with the classes. Another quantum-inspired supervised machine learning algorithm for multi-class classification based on so-called *pretty good measurement* has been proposed in [17], generalizing the Helstrom quantum state discrimination [18] that can be used for binary classification. Classification accuracy of this quantum-inspired multi-class classifier can be improved by increasing the number of copies of the quantum state that encodes the feature vector, at the cost of increasing the computational space and time.

In this paper, we analyze possible connections between support vector machines and quantum-inspired classifications using a geometric approach. In particular, considering a quantum encoding of classical data in terms of Bloch vectors of density operators, we observe that the execution of the Helstrom classifier is analogous to an SVM with a linear kernel. In Section 2, we give a short introduction to some quantum fundamentals that are relevant in the present work, such as the Bloch representation for quantum states. Moreover, we review the application of Helstrom state discrimination for binary classification. In Section 3, we analyze quantum state discrimination for binary classification encoding data into Bloch vectors, some empirical results in this regard and the code in Mathematica are presented in Appendix A. In particular, we highlight the SVM-like behavior of the Helstrom classification for a two-feature dataset considering the encoding in a bi-dimensional Hilbert space and an encoding into a space of enlarged dimension. In Section 4, we discuss the general strategy of implementing a nearest mean classifier based on an operator distance between quantum states such as trace distance and Bures distance. In Section 5, we present some numerical results obtained by the implementation of the Helstrom classifier. In Section 6, we draw some final comments.

## 2. Basics

The set of density matrices on the (finite-dimensional) Hilbert space $\mathsf{H}$ is given by $\mathfrak{S}(\mathsf{H}) = \{\rho \in \mathcal{B}^+(\mathsf{H}) : \mathrm{tr}\rho = 1\}$, where $\mathcal{B}^+(\mathsf{H})$ is the set of positive semidefinite operators on $\mathsf{H}$. The set $\mathfrak{S}(\mathsf{H})$ is convex and its extreme elements, the pure states, are rank-1 orthogonal projectors. A pure state has general form $\rho = |\psi\rangle\langle\psi|$, and it can then be directly identified with the unit vector $|\psi\rangle \in \mathsf{H}$ up to a phase factor.

The bases of the real space of Hermitian matrices on $\mathbb{C}^d$ can be used to decompose density matrices associated with states of a quantum system described in a $d$-dimensional Hilbert space. A fundamental basis for qubits (dim $\mathsf{H} = 2$) is formed by the three Pauli matrices and the $2 \times 2$ identity matrix. In this case, any density matrix can be represented by a three-dimensional vector, the *Bloch vector*, that lies within the unit ball in $\mathbb{R}^3$ whose boundary is the *Bloch sphere*. The points on the spherical surface are in bijective correspondence with the pure states. In higher dimensions, the set of quantum states is a convex

body with a much more complicated geometry and it is no longer simply represented as a unit ball. In general, for any $j, k, l$ such that $1 \leq j \leq d^2 - 1$, $0 \leq k < l \leq d - 1$, the *generalized Pauli matrices* $\sigma_j$ on $\mathbb{C}^d$ can be defined as follows:

$$
\sigma_j =: \begin{cases} \left| \frac{k}{d-1} \right\rangle \left\langle \frac{l}{d-1} \right| + \left| \frac{l}{d-1} \right\rangle \left\langle \frac{k}{d-1} \right| & \text{if } j \leq \frac{d(d-1)}{2} \text{ and } j = \frac{k(1-k)}{2} + (d-2)k + l; \\[2mm] -i \left| \frac{k}{d-1} \right\rangle \left\langle \frac{l}{d-1} \right| + i \left| \frac{l}{d-1} \right\rangle \left\langle \frac{k}{d-1} \right| & \text{if } \frac{d(d-1)}{2} < j \leq d(d-1) \text{ and } j = \frac{d(d-1)+k(1-k)}{2} + (d-2)k + l; \\[2mm] \sqrt{\frac{2}{l(l+1)}} \left( \sum_{k=0}^{l-1} \left| \frac{k}{d-1} \right\rangle \left\langle \frac{k}{d-1} \right| - l \left| \frac{l}{d-1} \right\rangle \left\langle \frac{l}{d-1} \right| \right) & \text{if } j > d(d-1) \text{ and } j = d(d-1) + l; \end{cases} \tag{1}
$$

where $\left\{ \left| \frac{k}{d-1} \right\rangle \right\}_{k=0,\dots,d-1}$ denotes the canonical basis of $\mathbb{C}^d$. The generalized Pauli matrices $\{\sigma_j\}_{j=1,\dots,d^2-1}$ are the standard generators of the special unitary group $SU(d)$. In particular, $\frac{d(d-1)}{2}$ matrices are symmetric, $\frac{d(d-1)}{2}$ matrices are antisymmetric, $d-1$ matrices are diagonal. Together with the $d \times d$ identity matrix $\mathtt{I}_d$, the generalized Pauli matrices form an orthogonal (the orthogonality is with respect to the Hilbert–Schmidt product $(A, B)_{HS} = \text{tr}(A^\dagger B)$) basis of the real space of $d \times d$ Hermitian matrices.

Let $\rho$ be a density operator on $\mathbb{C}^d$. The expansion of $\rho$ with respect to the orthogonal basis $\{\mathtt{I}_d, \sigma_j : 1 \leq j \leq d^2 - 1\}$ is:

$$
\rho = \frac{1}{d} \left( \mathtt{I}_d + \sqrt{\frac{d(d-1)}{2}} \sum_{j=1}^{d^2-1} b_j^{(\rho)} \sigma_j \right),
$$

where $b_j^{(\rho)} = \sqrt{\frac{d}{2(d-1)}} \text{tr}(\rho \, \sigma_j) \in \mathbb{R}$. The coordinates $\mathbf{b}^{(\rho)} = (b_1^{(\rho)}, \dots, b_{d^2-1}^{(\rho)})$ represent the Bloch vector associated to $\rho$ with respect to the basis $\{\mathtt{I}_d, \sigma_j : 1 \leq j \leq d^2 - 1\}$, which lies within the hypersphere of radius 1. For $d > 2$, the points contained in the unit hypersphere in $\mathbb{R}^{d^2-1}$ are not in bijective correspondence with quantum states on $\mathbb{C}^d$ such as in the case of a single qubit. However, any vector within the closed ball of radius $\frac{2}{d}$ gives rise to a density operator. From the physical viewpoint, the Bloch vector has real components that can be expressed as expectation values of measurable quantities. For $d = 3$, the generalized Pauli matrices are the Gell–Mann matrices and the Bloch vector can be expressed as expectation values of spin 1 operators.

A complex vector can be encoded into a density matrix. For instance, a quantum encoding (the *amplitude encoding*) is given by:

$$
\mathbb{C}^n \ni \mathbf{x} \mapsto |\mathbf{x}\rangle = \frac{1}{\sqrt{\| \mathbf{x} \|^2 + 1}} \left( \sum_{i=0}^{n-1} x_i |i\rangle + |n\rangle \right) \in \mathsf{H}, \tag{2}
$$

where $\{|i\rangle\}_{i=0,\dots,n}$ is the computational basis of the $(n+1)$-dimensional Hilbert space $\mathsf{H}$, identified as the standard basis of $\mathbb{C}^{n+1}$. The map defined in (2) encodes $\mathbf{x}$ into the pure state $\rho_{\mathbf{x}} = |\mathbf{x}\rangle\langle\mathbf{x}|$, the additional component of $|\mathbf{x}\rangle$ stores the norm of $\mathbf{x}$. Generally speaking, a *quantum encoding* is any procedure to encode classical information (e.g., a list of symbols) into quantum states. In this paper, we consider encodings of vectors in $\mathbb{C}^n$ and $\mathbb{R}^n$ into density matrices on a Hilbert space $\mathsf{H}$ whose dimension depends on $n$.

In [17], there is the proposal of a quantum-inspired classification algorithm based on a generalization of the Helstrom measurement, the so-called *Pretty Good measurement*, for quantum state discrimination. Let us focus on the case of binary classification of $n$-dimensional complex feature vectors; the algorithm is based on the following three ingredients: (1) a quantum encoding of the feature vectors $\mathbb{C}^n \ni \mathbf{x} \mapsto \rho_{\mathbf{x}} \in \mathfrak{S}(\mathsf{H})$; (2) the construction of the quantum centroids of the two classes $C_1$ and $C_2$ of training points:

$$
\rho_i := \frac{1}{|C_i|} \sum_{\mathbf{x} \in C_i} \rho_{\mathbf{x}} \qquad i = 1, 2; \tag{3}
$$

(3) application of the Helstrom discrimination on the two quantum centroids in order to assign a label to a new data instance.

Let us briefly introduce the notion of quantum state discrimination that is central in the present work. Given a set of arbitrary quantum states with respective a priori probabilities $R = \{(\rho_1, p_1), ..., (\rho_N, p_N)\}$, in general there is no measurement process that discriminates the states without errors. More formally, there does not exist a collection of effects $E = \{E_i\}_{i=1,...,N} \subset \mathcal{B}^+(\mathsf{H})$ such that $\sum_{i=1}^N E_i = \mathbb{I}$ satisfying the following property: $\text{tr}(E_i \rho_j) = 0$ when $i \neq j$ for all $i, j = 1, ..., N$. In some particular cases, the states can be exactly discriminated, for example if we have a set of orthogonal pure states $\{|\psi_1\rangle, ..., |\psi_N\rangle\}$, we can discriminate them without errors by means of the corresponding von Neumann measurement $\{|\psi_i\rangle\langle\psi_i|\}_{i=1,...,N}$. Returning to the general set $R$, the probability of a successful state discrimination performing the measurement $E$ is:

$$\mathbb{P}_E(R) = \sum_{i=1}^N p_i \text{tr}(E_i \rho_i). \tag{4}$$

An interesting and useful task is finding the optimal measurement that maximizes the probability (4). In [18], the author presents a complete characterization of the optimal measurement $E_{opt}$ for $R = \{(\rho_1, p_1), (\rho_2, p_2)\}$. $E_{opt}$ can be constructed as follows: Let $\Lambda := p_1\rho_1 - p_2\rho_2$ be the *Helstrom observable* whose positive and negative eigenvalues are, respectively, collected in the sets $D_+$ and $D_-$. Consider the two orthogonal projectors:

$$P_\pm := \sum_{\lambda \in D_\pm} P_\lambda, \tag{5}$$

where $P_\lambda$ projects onto the eigenspace of $\lambda$. The measurement $E_{opt} := \{P_+, P_-\}$ maximizes the probability (4) that attains the *Helstrom bound* $h_b(\rho_1, \rho_2) = p_1\text{tr}(P_+\rho_1) + p_2\text{tr}(P_-\rho_2)$.

Helstrom quantum state discrimination can be used to implement a binary classifier [17]. Let $\{(\mathbf{x}_1, y_1), ..., (\mathbf{x}_M, y_M)\}$ be a training set with $y_i \in \{1, 2\} \; \forall i = 1, ..., M$. Once a quantum encoding $\mathbb{C}^n \ni \mathbf{x} \mapsto \rho_\mathbf{x} \in \mathfrak{S}(\mathsf{H})$ has been selected, one can construct the quantum centroids $\rho_1$ and $\rho_2$ as in (3) of the two classes $C_{1,2} = \{\mathbf{x}_i : y_i = 1, 2\}$. Let $\{P_+, P_-\}$ be the Helstrom measurement defined by the set $R = \{(\rho_1, p_1), (\rho_2, p_2)\}$, where the probabilities attached to the centroids are $p_{1,2} = \frac{|C_{1,2}|}{|C_1| + |C_2|}$. The *Helstrom classifier* applies the optimal measurement for the discrimination of the two quantum centroids to assign the label $y$ to a new data instance $\mathbf{x}$, encoded into the state $\rho_\mathbf{x}$, as follows:

$$y(\mathbf{x}) = \begin{cases} 1 & \text{if} \quad \text{tr}(P_+\rho_\mathbf{x}) \geq \text{tr}(P_-\rho_\mathbf{x}) \\ 2 & \text{otherwise} \end{cases} \tag{6}$$

A strategy to increase the accuracy in classification is given by the construction of the tensor product of $k$ copies of the quantum centroids $\rho_{1,2}^{\otimes k}$ enlarging the Hilbert space where data are encoded. The corresponding Helstrom measurement is $\{P_+^{\otimes k}, P_-^{\otimes k}\}$, and the Helstrom bound satisfies [17]:

$$h_b(\rho_1^{\otimes k}, \rho_2^{\otimes k}) \leq h_b\left(\rho_1^{\otimes(k+1)}, \rho_2^{\otimes(k+1)}\right) \qquad \forall k \in \mathbb{N}. \tag{7}$$

Enlarging the Hilbert space of the quantum encoding, one increases the Helstrom bound obtaining a more accurate classifier. The computational cost is evident; however, in the next section, we observe that in the case of real input vectors, the space can be enlarged, saving time and space by means of the encoding into Bloch vectors.

## 3. Geometric Approach to Quantum-Inspired Classifications

In [17], a real vector $\mathbf{x} \in \mathbb{R}^{d-1}$ is encoded, as shown above, in a projector operator $\rho_\mathbf{x}$ of a Hilbert space $\mathbb{C}^d$ represented by an $d \times d$ real symmetric matrix, where $d \geq 2$. For simplicity, we consider an input vector $[x_1, x_2] \in \mathbb{R}^2$ and the corresponding projector

operator $\rho_{[x_1,x_2]}$ on $\mathbb{C}^3$. By easy computations, one can see that the Bloch vector of $\rho_{[x_1,x_2]}$ has null components:

$$\mathbf{b}^{(x_1,x_2)} = \frac{1}{1+x_1^2+x_2^2}\left[2x_1x_2, 2x_1, 2x_2, 0, 0, 0, x_1^2 - x_2^2, \frac{x_1^2 + x_2^2 - 2}{\sqrt{3}}\right]. \tag{8}$$

Instead of using a matrix with nine real elements, memory occupation can be improved by considering only the non-zero components of the Bloch vector. In general, the technique of removing the components that are zero or repeated several times allows reducing the space and the calculation time considering only the significant values that allow to carry out the classification.

Quantum-inspired classifications are similar to support vector machines that implicitly map the input space into high-dimensional feature space using kernel functions, where the maximal separating margins are constructed. In this case, the nonlinear explicit injective function $\varphi : \mathbb{R}^2 \to \mathbb{R}^5$ can be defined as follows:

$$\varphi([x_1, x_2]) := \frac{1}{x_1^2 + x_2^2 + 1}\left[2x_1x_2, 2x_1, 2x_2, x_1^2 - x_2^2, \frac{x_1^2 + x_2^2 - 2}{\sqrt{3}}\right]. \tag{9}$$

From a geometric point of view, feature vectors are indeed points on the surface of a hyper-hemisphere. The corresponding elements to the quantum centroids $\rho_1, \rho_2$ are the centroids of the feature vectors:

$$\bar{\mathbf{x}}_i := \frac{1}{|C_i|} \sum_{\mathbf{x} \in C_i} \varphi(\mathbf{x}) \qquad i = 1, 2 \tag{10}$$

In general, such centroids are points inside the hypersphere and therefore they do not have an inverse image.

The Helstrom classifier can also be applied in a smaller space using the following encoder from $\mathbb{R}^2$ to density operators of $\mathbb{C}^2$:

$$\rho_{[x_1,x_2]} = \frac{1}{2}\left(\mathbb{I}_2 + \sum_{j=1}^{3} b_j \sigma_j\right), \tag{11}$$

where the Bloch vector $\mathbf{b} = \varphi([x_1, x_2]) \in \mathbb{R}^3$ and $\varphi([x_1, x_2]) := \frac{1}{\sqrt{x_1^2+x_2^2+1}}[x_1, x_2, 1]$. As discussed in Section 5, the Helstrom classifier gives even less accurate results on the training set as expected because the feature space is smaller than the previous one. In this particular case, quantum centroids are points inside the Bloch sphere of a qubit that correspond to density operators.

An interesting question suggested in [17] is whether classification accuracy can be improved by increasing the dimension of the state space of density matrices that represent input vectors. Improving accuracy providing $n$ copies of centroids in quantum-inspired classifications has a strong impact in terms of computational space (from dimension $d - 1$ to $d^{2n}$) and time. Following the geometric approach, considering the significant values that allow to carry out the classification, the explicit function $\varphi : \mathbb{R}^2 \to \mathbb{R}^{20}$ for two copies can be defined as follows:

$$\varphi([x_1, x_2]) := \frac{1}{(x_1^2 + x_2^2 + 1)^2}\Big[2x_1^3x_2, 2x_1^3, 2x_1^2x_2^2, 2x_1^2x_2, 2x_1^2, 2x_1x_2^3, 2x_1x_2^2, 2x_1x_2, 2x_1, 2x_2^3,$$

$$2x_2^2, 2x_2, x_1^2(x_1 - x_2)(x_1 + x_2), \frac{x_1^2(x_1^2 + x_2^2 - 2)}{\sqrt{3}}, \frac{x_1^2(x_1^2 - 2x_2^2 + 1)}{\sqrt{6}},$$

$$\frac{x_1^4 - 4x_2^4 + x_1^2(2x_2^2 + 1)}{\sqrt{10}}, \frac{x_1^2 + x_1^4 - 5x_2^2 + 2x_1^2x_2^2 + x_2^4}{\sqrt{15}}, \frac{x_1^4 + x_2^2 + x_2^4 + x_1^2(2x_2^2 - 5)}{\sqrt{21}},$$

$$\frac{x_1^4 - 6x_2^2 + x_2^4 + 2x_1^2(x_2^2 + 1)}{2\sqrt{7}}, \frac{1}{6}(x_1^2 + x_2^2 - 2)(x_1^2 + x_2^2 + 4)\Big]. \tag{12}$$

In particular, removing null and multiple entries, we consider only 20 values instead of 81 for two copies, 51 values instead of 729 for three copies and so on. However, one must also take into account high-precision numbers and track the propagation of the numerical error. The gain in accuracy seems marginal already from three copies.

In Section 5, we will show some numerical results obtained by the implementation of the Helstrom classifier and, in particular, the close analogy between this quantum-inspired method and the SVM with linear kernel. A natural limitation of Helstrom classification arises in the case of a training set where the centroids of the two classes coincide, the Helstrom classifier is clearly useless because it is not able to perform a corresponding state discrimination. In the same situation, there are effective classical classification methods such as *Random Forest*, *Naive Bayes classifier* and *Nearest Neighbor*.

## 4. Quantum-Inspired Nearest Mean Classifications

In [16], a quantum version of the nearest mean classifier was shown making use of the inverse of the stereographic projection as encoder:

$$\pi^{-1} : \mathbb{R}^{d-1} \ni \mathbf{x} \mapsto \frac{2}{\sum_{i=1}^{d-1} x_i^2 + 1}\Big[x_1, \ldots, x_{d-1}, \frac{\sum_{i=1}^{d-1} x_i^2 - 1}{2}\Big] \in \mathbb{S}^{d-1}. \tag{13}$$

In the case $\mathbf{x} \in \mathbb{R}^2$, the lowest dimensional quantum encoding is obviously in $\mathbb{C}^2$; in particular, $\rho_{\mathbf{x}}$ is given by the density matrix identified by the Bloch vector $\pi^{-1}(\mathbf{x})$, that is:

$$\rho_{\mathbf{x}} = \frac{1}{x_1^2 + x_2^2 + 1}\begin{pmatrix} x_1^2 + x_2^2 & x_1 - ix_2 \\ x_1 + ix_2 & 1 \end{pmatrix}. \tag{14}$$

The state $\rho_{\mathbf{x}}$ is pure, i.e., a projector, as $\pi^{-1}(\mathbf{x})$ lies on the surface of the Bloch sphere. In Appendix A, the encoding $\mathbf{x} \mapsto \rho_{\mathbf{x}}$ defined by (14), in an arbitrary dimension, is realized by the function $SVMEncoder[\mathbf{x}, type]$ with $type = 2$. For binary classification, in [16], the centroids of the two classes are calculated in the feature space and then encoded into density matrices according to (14). Given a test point encoded into a density matrix, the classifier appends it to the nearest centroid with respect to the normalized trace distance:

$$\overline{d}_{tr}(\rho_{\mathbf{x}}, \rho_{\mathbf{y}}) = \frac{2}{\sqrt{(1 - b_{\mathbf{x}_3})(1 - b_{\mathbf{y}_3})}} d_{tr}(\rho_{\mathbf{x}}, \rho_{\mathbf{y}}), \tag{15}$$

where $b_{\mathbf{x}_3}$ and $b_{\mathbf{y}_3}$ are the Bloch coefficients with respect to the Pauli matrix $\sigma_3$ of $\rho_{\mathbf{x}}$ and $\rho_{\mathbf{y}}$ and $d_{tr}(\rho_{\mathbf{x}}, \rho_{\mathbf{y}}) = \frac{1}{2}\text{tr}(|\rho_{\mathbf{x}} - \rho_{\mathbf{y}}|)$. One can easily verify that:

$$\overline{d}_{tr}(\rho_{\mathbf{x}}, \rho_{\mathbf{y}}) = d_E(\mathbf{x}, \mathbf{y}), \tag{16}$$

where $d_E$ is the standard Euclidean distance. Therefore a quantum-inspired nearest mean classifier can be defined by the encoding given in (14) and by the evaluation of the trace distance between density operators. In [16], experimental results on the performances of such a quantum-inspired classifier are presented and compared to the classical nearest mean classifier with impressive results in terms of accuracy.

As illustrated in the previous section, in order to improve data separation, the input space can be mapped into a higher dimensional feature space by means of a *kernel trick*. It is also possible to reduce the computational space from $\mathbb{R}^8$ to $\mathbb{R}^5$ in this case with the following explicit function $\varphi : \mathbb{R}^2 \to \mathbb{R}^5$:

$$\varphi([x_1, x_2]) := \frac{2}{(x_1^2 + x_2^2 + 1)^2}[4x_1x_2, 2x_1(x_1^2 + x_2^2 - 1), 2x_2(x_1^2 + x_2^2 - 1),$$

$$2x_1^2 - 2x_2^2, \frac{4x_1^2 + 4x_2^2 - x_1^4 - x_2^4 - 2x_1^2 x_2^2 - 1}{\sqrt{3}}]. \tag{17}$$

In Appendix A, we implement the higher dimensional encoding induced by $\varphi$ calling the function *BlochVector*[*SVMEncoder*[$\mathbf{x}$, 2]] and removing the null components.

The following distances, respectively, the Hilbert–Schimidt distance, trace distance, Bures distance and Hellinger distance, are often considered and can be used for nearest mean classification:

$$d_{HS}(\rho_1, \rho_2) = \sqrt{\mathrm{tr}|\rho_1 - \rho_2|^2},$$

$$d_{\mathrm{tr}}(\rho_1, \rho_2) = \frac{1}{2}\mathrm{tr}|\rho_1 - \rho_2|,$$

$$d_B(\rho_1, \rho_2) = \sqrt{2 - 2\,\mathrm{tr}\sqrt{\sqrt{\rho_1}\rho_2\sqrt{\rho_1}}},$$

$$d_{He}(\rho_1, \rho_2) = \sqrt{2 - 2\,\mathrm{tr}\sqrt{\rho_1}\sqrt{\rho_2}},$$

where $|A| = \sqrt{A^\dagger A}$ is the modulus of the operator $A$. The measures induce different geometries. The set of states of a qubit is equivalent to the Bloch sphere for the Hilbert–Schmidt distance $d_{HS}$ and for the trace distance $d_{\mathrm{tr}}$, and to the Uhlmann hemisphere for the Bures distance $d_B$. For higher dimensions, the geometries induced by the Hilbert–Schmidt distance and the trace distance also differ. The Bures distance and trace distance are useful measures for quantifying the states distinguishability. The Bures distance is an optimized Kullback–Leibler distance between output statistics over all quantum measurements. The trace distance is a function of the probability to successfully discriminate two states in a single measurement optimized over all quantum measurements. As mentioned above, in the bi-dimensional case there is the equivalence of the normalized trace distance and the Euclidean distance (16). Moreover, one can see that the trace distance between pure states is equal to half of the Euclidean distance between the respective Bloch vectors. Therefore, in the Mathematica code of Appendix B, the function *CentroidClassify* for nearest mean classification based on trace distance is equivalently defined by means of the standard norm.

Within the paradigm of quantum-inspired classification, the Bures distance and the Hellinger distance can be used to define other classifiers that evaluate these distances for the nearest mean classification. Let us consider a binary classification problem and the quantum centroids (3) of the two classes. We can define a classification algorithm that evaluates the Bures distance between the pure quantum state encoding a test point and the quantum centroids that are not pure in general. The fidelity between density operators, defined as $\mathcal{F}(\rho_1, \rho_2) = \left(\mathrm{tr}\sqrt{\sqrt{\rho_1}\rho_2\sqrt{\rho_1}}\right)^2$, reduces to $\mathcal{F}(\rho_1, \rho_2) = \langle\psi_1|\rho_2|\psi_1\rangle$ when $\rho_1 = |\psi_1\rangle\langle\psi_1|$. Therefore, the Bures distance between the pure state $\rho_{\hat{\mathbf{x}}}$ encoding the test point $\hat{\mathbf{x}}$ and the quantum centroid $\rho_i$ is:

$$d_B(\rho_{\mathbf{x}}, \rho_i) = \sqrt{2 - 2\sqrt{\frac{1}{d}\left(1 + (d-1)\mathbf{b}^{(\mathbf{x})}\cdot\mathbf{b}^{(i)}\right)}} \equiv D_B\left(\mathbf{b}^{(\mathbf{x})}, \mathbf{b}^{(i)}\right), \tag{18}$$

where $\mathbf{b}^{(\mathbf{x})}$ and $\mathbf{b}^{(i)}$ are the Bloch vectors of $\rho_{\mathbf{x}}$ and $\rho_i$, respectively, and $d$ is the dimension of the Hilbert space of the quantum encoding. The formula (18) can be directly derived from $\mathrm{tr}(\rho_1\rho_2) = \frac{1}{d}(1 + (d-1)\mathbf{b}^{(1)}\cdot\mathbf{b}^{(2)})$, which is an immediate consequence of the fact that the generalized Pauli matrices are traceless and satisfy $\mathrm{tr}(\sigma_i\sigma_j) = 2\delta_{ij}$. An example of the nearest mean classifier based on the Bures distance can be defined by Algorithm 1.

The quantum encodings at line 1 and line 3 of Algorithm 1 can be realized by the function *SVMEncoder*[$\mathbf{x}$, 1], defined in Appendix A, for instance. At line 5, the quantum centroids are constructed according to (3); alternatively, the classifier can calculate the centroids in the feature space from the Bloch vectors of the quantum states encoding the

training points like in (10), where in general, the resulting centroid vectors $\bar{\mathbf{x}}_i$ are not the Bloch vectors of density operators.

---

**Algorithm 1:** *Quantum-inspired nearest mean classifier based on Bures distance.*

---

    **Input** : Two classes $C_1$ and $C_2$ of training points, unlabelled point $\hat{\mathbf{x}}$
    **Result** : Label $y$ of $\hat{\mathbf{x}}$

1   encode $\hat{\mathbf{x}}$ into a pure state $\rho_{\hat{\mathbf{x}}}$;
2   **foreach** $x \in C_1$ *or* $x \in C_2$ **do**
3     |   encode $\mathbf{x}$ into a pure state $\rho_{\mathbf{x}}$;
4   **end**
5   construct the quantum centroids $\rho_i = \frac{1}{|C_i|} \sum_{\mathbf{x} \in C_i} \rho_{\mathbf{x}}$ for $i = 1, 2$ ;
6   $\mathbf{b}^{(\hat{\mathbf{x}})} \leftarrow BlochVector[\rho_{\hat{\mathbf{x}}}]$;
7   $\mathbf{b}^{(i)} \leftarrow BlochVector[\rho_i]$ for $i = 1, 2$;
8   **if** $D_B\left(\mathbf{b}^{(\hat{\mathbf{x}})}, \mathbf{b}^{(1)}\right) \leq D_B\left(\mathbf{b}^{(\hat{\mathbf{x}})}, \mathbf{b}^{(2)}\right)$ **then**
9     |   **return** $y = 1$
10  **else**
11    |   **return** $y = 2$
12  **end**

---

## 5. Numerical Results and Discussion

We focus on some numerical results obtained running the considered quantum-inspired classifiers on some datasets. In Appendix A, there is the Mathematica code of the tests on the Helstrom classifier, and in Appendix B, there is the Mathematica code of the implementation of the quantum-inspired NMC (the code is also available at the following repository: github.com/leporini/classification). As a benchmark for testing the Helstrom classifier, we applied the DB-SCAN clustering algorithm [19] to a *moons dataset*, obtaining the classification of Figure 1.

The first test provides the quantum encoding of the bi-dimensional input vectors into a five-dimensional space according to (9) and (10) and the execution of the Helstrom classifier on the moons dataset. The obtained decision boundary and the misclassified points are shown in Figure 2. The same classification task has been tackled by a classical SVM with a linear kernel, which returns the decision line and the support vectors depicted in Figure 3. The comparison of the outputs reveals that the execution of the Helstrom classifier returns the decision boundary of an SVM with the linear kernel.

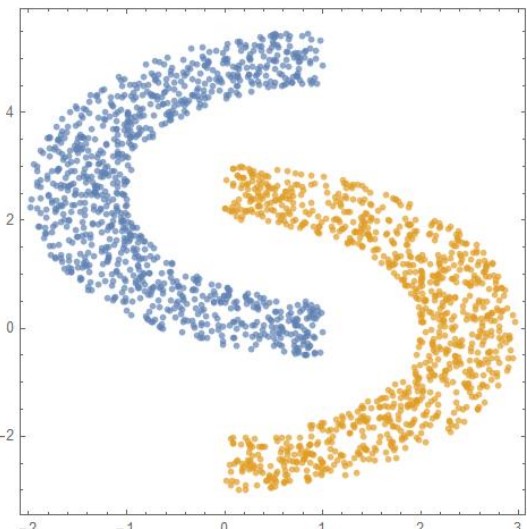

**Figure 1.** DBSCAN classification.

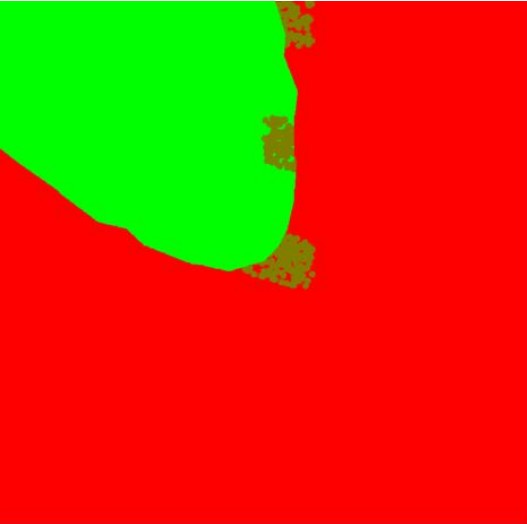

**Figure 2.** Helstrom classification with highlighted points classified incorrectly.

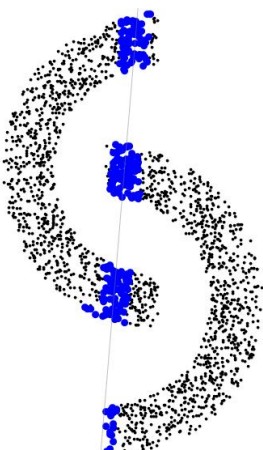

**Figure 3.** SVM with linear kernel behaves similarly to the Helstrom method.

The considered test on the performance of the Helstrom classifier over the moons dataset can be repeated considering a smaller space encoding the data points into density operators of a qubit according to (11). Since we are considering a feature map $\varphi : \mathbb{R}^2 \to \mathbb{R}^3$, which represents data in a lower dimensional space, the classifier is less accurate as expected. The accuracy of the Helstrom classifier, over the training set, in the two cases is:

$$Acc_{\mathbb{R}^5}^{Hel} = 0.8455, \qquad Acc_{\mathbb{R}^3}^{Hel} = 0.503. \tag{19}$$

Accuracy is sensitive to the dimension, as shown in Figure 4.

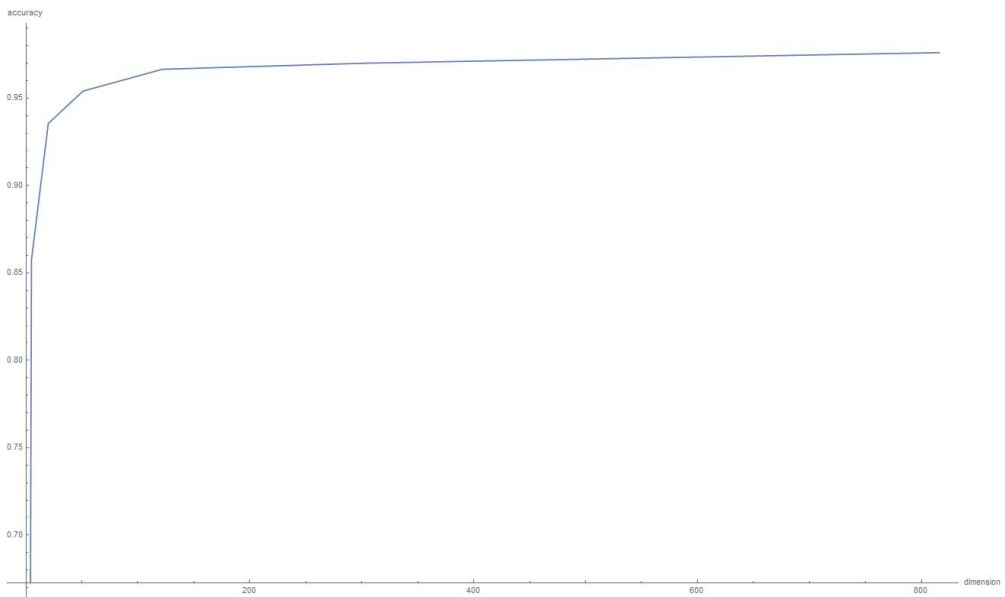

**Figure 4.** Accuracy of the Helstrom classifier as a function of the dimension of the space.

In Figure 5, there is the decision boundary found by the Helstrom classifier, with misclassified points, in the lowest dimensional case. Figure 6 shows the output of an SVM with a linear kernel, and its behavior is confirmed as similar to the Helstrom classifier.

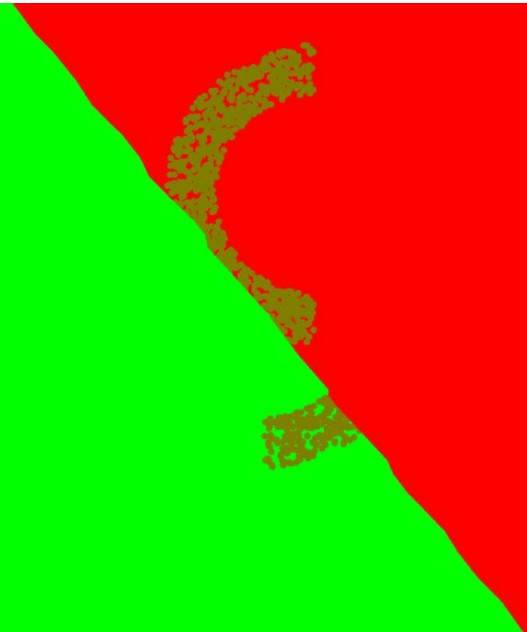

**Figure 5.** Helstrom method applied to the smallest space with highlighted points classified incorrectly.

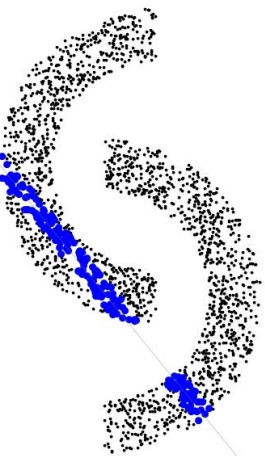

**Figure 6.** SVM with linear kernel behaves similarly to the Helstrom method in the smallest space.

In the case of a training set where the centroids of the two classes coincide, the Helstrom classifier does not work because it is not able to perform a state discrimination. Let us consider the dataset represented in Figure 7 with coinciding centroids; on the one hand, the Hesltrom classifier is useless, and on the other hand, RandomForest, Naive Bayes classifier, and Nearest Neighbors present the following high accuracies on the training set:

$$Acc^{RF} = 0.975, \qquad Acc^{NB} = 0.955, \qquad Acc^{NN} = 0.985. \tag{20}$$

Considering a dataset with distinguishable but close centroids, the performance of the Helstrom classifier is poor with respect to existing classical methods. For example, let us consider the training set represented in Figure 8. The accuracies of the Helstrom classifier, the Random Forest and the Nearest Neighbors can be compared, observing that the performances of the classical algorithms are definitely better in terms of the correct classification:

$$Acc^{Hel} = 0.73, \qquad Acc^{RF} = 0.98, \qquad Acc^{NN} = 0.85. \tag{21}$$

In Figure 9, there are the misclassified points by the Helstrom classifier.

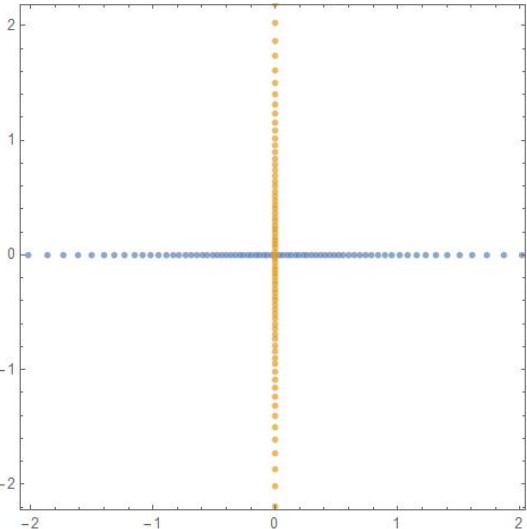

**Figure 7.** Helstrom classification is useless with the same centroids, while some classical methods work.

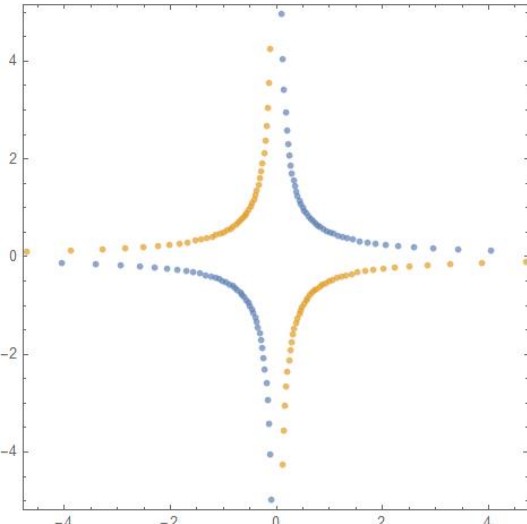

**Figure 8.** Helstrom classification is useless with centroids close to each other, while some classical methods work.

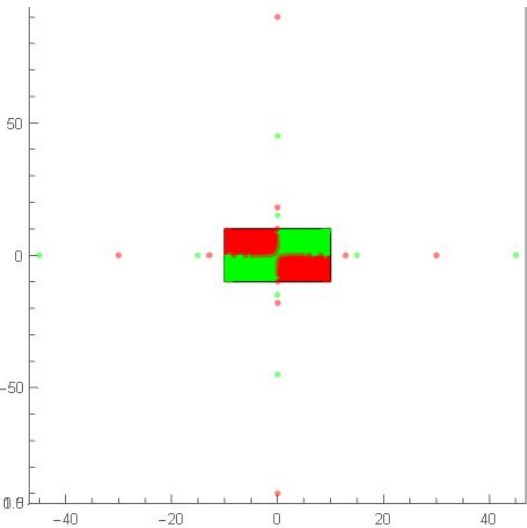

**Figure 9.** Helstrom method with highlighted points classified incorrectly.

The numerical results presented in this section reveal the connection between the Helstrom classifier and SVM with linear kernel; in particular, one can observe that different strategies of quantum encoding by means of Bloch vectors of density operators correspond to different kernel tricks.

In Section 4, we showed that from a geometric viewpoint, a quantum-inspired nearest mean classifier based on the trace distance between density operators can be equivalently implemented considering the Euclidean distance between Bloch vectors. We considered both the implementations of this quantum-inspired NMC, finding the exact equivalence in terms of accuracy on the moons dataset:

$$Acc^{Density} = Acc^{Bloch} = 0.865. \tag{22}$$

$Acc^{Density}$ is the accuracy of the NMC based on the quantum encoding of data points into density operators according to (14) and the calculation of trace distances within this representation. $Acc^{Bloch}$ is the accuracy of the NMC, which encodes the data points into Bloch vectors by means of the inverse of the stereographic projection and evaluates the Euclidean distance in the *Bloch feature space*.

## 6. Conclusions

In this paper, we analyzed some methods of quantum-inspired classification, highlighting a connection with support vector machines. After an introduction on the Bloch representation of quantum states in an arbitrary dimension, we considered the Helstrom quantum state discrimination applied to binary classification (the *Helstrom classifier*), observing that its execution is similar to an SVM with linear kernel. In particular, adopting a geometric viewpoint, we described how quantum encodings of feature vectors can be used to implement a kernel trick, improving the quality of classification. Moreover, if one considers multiple copies of the encoding quantum states to map real feature vectors into a space with higher dimensions (as performed in [6] to obtain a polynomial kernel for a quantum SVM with an exponential cost of space resources), we showed that the computational cost can be calmed down, deleting the redundancies in the resulting Bloch vectors. In this way, in quantum-inspired classification, one can define a nonlinear injective function to perform a kernel trick saving space and time.

We presented some experimental results on the moons dataset to exhibit the behavior of the Helstrom classifier as an SVM with a linear kernel. With this dataset, we have enlarged the dimension of the Hilbert space, changing the quantum encoding within the Bloch representation of the density operators. On the other hand, we gave a couple of examples where the Helstrom classifier does not work due to the difficult discrimination of the quantum states representing the centroids of the classes.

We also focused on quantum-inspired nearest mean classification that is based on the computation of operator distances between density matrices encodings feature vectors. For instance, the classifier can evaluate the trace distance or the Bures distance among encoding quantum states. We considered the classifier with trace distance showing that it is equivalent (in terms of classification accuracy) to the "geometric" classifier, which evaluates the Euclidean distance between Bloch vectors. Then, we proposed an algorithm based on Bures distance, which can be evaluated directly in terms of Bloch vectors. An empirical study on this quantum-inspired classifier is a matter for future works.

As a general consideration emerging from the present work, we point out that the geometric approach considering the Bloch representation of density matrices is suitable to describe quantum-inspired classification. This approach reveals a connection between the Helstrom classifier based on quantum state discrimination and SVM. The geometric viewpoint seems to be fruitful also to define quantum-inspired nearest mean classifiers.

The present work opens possible directions of investigation such as the full characterization of the kernel of the Helstrom classifier in order to complete the description of quantum state discrimination as the execution of a support vector machine. More in general, an interesting topic could be a satisfying geometric analysis of quantum-inspired machine learning algorithms beyond classifiers since the present paper suggests that the geometry of quantum states offers a novel machinery to deal with data.

**Author Contributions:** Conceptualization, R.L. and D.P.; software, R.L.; validation, D.P.; formal analysis, D.P.; writing—original draft preparation, R.L. and D.P.; writing—review and editing, D.P. All authors have read and agreed to the published version of the manuscript.

**Funding:** This research was funded by Q@TN consortium.

**Institutional Review Board Statement:** Not applicable.

**Informed Consent Statement:** Not applicable.

**Data Availability Statement:** The code is also available at the following repository: github.com/leporini/classification (accessed on 28 August 2021).

**Conflicts of Interest:** The authors declare no conflict of interest.

## Appendix A. Mathematica Notebook for Quantum-Inspired Classifications

Needs["QDENSITYQdensity", "C:\\Qdensity.m"];

KetBra::usage = "$|\frac{x}{d-1}\rangle\langle\frac{y}{d-1}|$ of $\mathbb{C}^d$, where x,y$\in$ {0,...,d-1} and $|0\rangle, |\frac{1}{d-1}\rangle, ..., |1\rangle$ are the elements of the canonical basis of $\mathbb{C}^d$ ";

KetBra[d_, x_, y_]:=SparseArray[$\{\{x+1, y+1\} \to 1\}, \{d, d\}$];

BlochVector::usage = "BlochVector[$\rho$], where $\rho$ is a density operator of $\mathbb{C}^d$ and d$\geq$2.

\n\t Returns the corresponding Bloch vector b$\in R^{d^2-1}$.";

BlochVector[$\rho$_]:=Block[{$d$, sigma},

$d$ = Dimensions[$\rho$][[2]];

sigma = {};

Do[AppendTo[sigma, KetBra[$d, l, k$] + KetBra[$d, k, l$]], {$l, 0, d-2$}, {$k, l+1, d-1$}];

Do[AppendTo[sigma, $-i$KetBra[$d, k, l$] + $i$KetBra[$d, l, k$]], {$l, 0, d-2$}, {$k, l+1, d-1$}];

Do$\left[$AppendTo$\left[$sigma, $\sqrt{\frac{2}{l(l+1)}}$ (Sum[KetBra[$d, k, k$], {$k, 0, l-1$}] $- l$KetBra[$d, l, l$])$\right]$, {$l, 1, d-1$}$\right]$;

Tr[$\rho$.#]&/@sigma];

BlochVectorInverse::usage = "BlochVectorInverse[b]

\n\t Returns the density operator of $\mathbb{C}^d$ of the Bloch vector b$\in R^{d^2-1}$.

\n\t Not any vector b of the unit hypershpere gives rise a density operator,

since the output is not a semi-definite positive operator (i.e. there exists a negative eigenvalue),

\n\t but all vectors of length $\leqslant \frac{2}{d}$ give rise a density operator.";

BlochVectorInverse[b_]:=Block[{$d$, sigma, $\rho$},

$d$ = Ceiling$\left[\sqrt{\text{Length}[b]+1}\right]$;

sigma = {};

Do[AppendTo[sigma, KetBra[$d, l, k$] + KetBra[$d, k, l$]], {$l, 0, d-2$}, {$k, l+1, d-1$}];

Do[AppendTo[sigma, $-i$KetBra[$d, k, l$] + $i$KetBra[$d, l, k$]], {$l, 0, d-2$}, {$k, l+1, d-1$}];

Do$\left[$AppendTo$\left[$sigma, $\sqrt{\frac{2}{l(l+1)}}$ (Sum[KetBra[$d, k, k$], {$k, 0, l-1$}] $- l$KetBra[$d, l, l$])$\right]$, {$l, 1, d-1$}$\right]$;

$b = \frac{2}{d}$PadRight[$b, d$];

$\rho = \frac{1}{d}$IdentityMatrix[$d$] $+ \sqrt{\frac{d-1}{2d}}$Sum[$b[[j]]$sigma[[$j$]], {$j, 1, d^2-1$}];

Return[$\rho$]];

SVMEncoder::usage = "SVMEncoder[x,type], where x$\in R^{d-1}$ (keeps the value |x| and normalizes the new vector if type=1).

\n\t When type=2, it uses the inverse of the stereographic projection.

\n\t Returns the density operator of $\mathbb{C}^d$,

which is projection-operator (that projects over the closed subspace determined by the normalized vector) if type is 1 or 2.

\n\t It is a mixed state with the corresponding Bloch vector of length $\frac{2}{d}$ if type=3.";

SVMEncoder[x_, type_]:=Block[{$u, \rho$},

$u$ = Switch$\left[$type, 1, Normalize[Append[$x, 1$]], 2, $\frac{2}{\text{Total}[x^2]+1}$Append$\left[x, \frac{1}{2}(\text{Total}[x^2]-1)\right]$, 3, Normalize[Append[$x, 1$]]$\right]$;

If[type == 3, $\rho$ = BlochVectorInverse[$u$], $\rho$ = Outer[Times, $u, u$]]; Return[$\rho$]];

BinaryClassifier::usage = "BinaryClassifier[DClass1,DClass2],

where DClass1 and DClass2 are density operators of the class 1 and 2, respectively.

\n\t Returns the projection-operators P+ and P-.";

BinaryClassifier[DClass1_, DClass2_]:=Block[{$p, c$, vals, vecs, prjs, prj1, prj2},

$p = \frac{\text{Length}[\text{DClass1}]}{\text{Length}[\text{DClass1}]+\text{Length}[\text{DClass2}]}$;

$c = p$Total[DClass1] $- (1-p)$Total[DClass2]; {vals, vecs} = Eigensystem[$c$];

vecs = Normalize/@vecs; prjs = Outer[Times, Conjugate[#], #]&/@vecs;

prj1 = Total[Pick[prjs, NonNegative[vals]]];

prj2 = Total[Pick[prjs, Negative[vals]]];

Return[{prj1, prj2}]];

HelstromClassify::usage = "HelstromClassify[{P+,P-},$\rho$].

\n\t Returns the class 1 or 2.";

HelstromClassify[{prj1_, prj2_}, $\rho$_]:=If[Tr[prj1.#] $\geq$ Tr[prj2.#], 1, 2]&/@$\rho$;

ToDensity[B_]:=Block[{sigma, $\rho$},

sigma = {KetBra[2, 0, 1] + KetBra[2, 1, 0], $-i$KetBra[2, 1, 0] + $i$KetBra[2, 0, 1], KetBra[2, 0, 0] $-$ KetBra[2, 1, 1]};

```
ρ = {}; Do[AppendTo[ρ, ½ (IdentityMatrix[2] + Sum[B[[i, j]]sigma[[j]], {j, 3}])], {i, Length[B]}]; Return[ρ]];
unstandardize[point_, sd_, mu_]:=point * sd + mu;
getvectors[cf_ClassifierFunction, X_]:=With[{p = X}, With[{sd = StandardDeviation[p], mu = Mean[p]},
unstandardize[#, sd, mu]&/@cf[[1]]["Model"]["TrainedModel"][[1]]["supportVectors"]]]
getplane[svmcf_ClassifierFunction, X_]:=With[{tm = svmcf[[1]]["Model"]["TrainedModel"][[1]], points = X},
Module[{sv = tm["supportVectors"], svc = tm["supportVectorCoefficients"],
sd = StandardDeviation[points], mu = Mean[points], dim = Length[points[[1]]], vecs, offset},
vecs = Rest[RotationMatrix[{svc.sv, PadRight[{1}, dim]}].IdentityMatrix[dim]];
offset = With[{vars = Array[x, dim]}, Values@First@FindInstance[vars.(svc.sv) == tm["rho"], vars]];
vecs = sd * #&/@vecs;
offset = unstandardize[offset, sd, mu];
If[dim > 2, InfinitePlane[offset, vecs], InfiniteLine[offset, First[vecs]]]]]]
        DBSCAN with a toy dataset
circle[r_, theta_]:={rSin[theta], rCos[theta]};
{train, test} = With[{rot = RotationTransform[π, {0, 0}], tra = TranslationTransform[{1, 2.5}],
pts = circle@@@RandomVariate[UniformDistribution[{{2, 3}, {0, Pi}}], 2000]},
TakeDrop[RandomSample@Join[tra[rot[pts]], pts], 2000]];
cl = ClusterClassify[train, Method → "DBSCAN"]
ListPlot[Pick[test, cl[test], #]&/@Range[2], PlotStyle → Directive[PointSize[0.013], Opacity[0.7]],
AspectRatio → 1, Frame → True, Axes → False]
{X1, X2} = Pick[train, cl[train], #]&/@Range[2];
{X1Test, X2Test} = Pick[test, cl[test], #]&/@Range[2];

D1 = SVMEncoder[#, 1]&/@X1;
D2 = SVMEncoder[#, 1]&/@X2;
{p1, p2} = BinaryClassifier[D1, D2];
y1 = HelstromClassify[{p1, p2}, D1];
y2 = HelstromClassify[{p1, p2}, D2];
accuracy[y1_, y2_]:=N[ (Count[y1,1]+Count[y2,2]) / (Length[y1]+Length[y2]) ];
accuracy[y1, y2]
Show[Plot3D[{Tr[p1.SVMEncoder[{x1, x2}, 1]], Tr[p2.SVMEncoder[{x1, x2}, 1]]}, {x1, −10, 10}, {x2, −10, 10},
PlotStyle → {Green, Red}, ViewPoint → {0, 0, ∞}, Lighting → {{"Ambient", White}}, Mesh → False],
Graphics3D[{RGBColor[0, 1, 0, 0.5], PointSize[0.013], Point[Append[#, 1]&/@X1]}],
Graphics3D[{RGBColor[1, 0, 0, 0.5], PointSize[0.013], Point[Append[#, 1]&/@X2]}]]
y = Join[y1, y2];
X = Join[X1, X2];
HelstromX1 = Pick[X, y, 1];
HelstromX2 = Pick[X, y, 2];
svm = Classify[X → y, Method → {"SupportVectorMachine", "KernelType" → "Linear"}];
Graphics[{Point[X], Blue, PointSize[Large], Point[getvectors[svm, X]], Opacity[0.5], Gray, getplane[svm, X]}]
AssociationMap[ClassifierMeasurements[Classify[X → y, Method → {"SupportVectorMachine", "KernelType" → #}],
1 → HelstromX1, 2 → HelstromX2, "Accuracy"]&, {"Linear", "RadialBasisFunction", "Polynomial", "Sigmoid"}]
D1Test = SVMEncoder[#, 1]&/@X1Test;
D2Test = SVMEncoder[#, 1]&/@X2Test;
y1 = HelstromClassify[{p1, p2}, D1Test];
y2 = HelstromClassify[{p1, p2}, D2Test];
accuracy[y1, y2]
```

0.8455 is the Helstrom accuracy on the training set

Linear → 0.9475, RadialBasisFunction → 0.9915, Polynomial → 0.996, Sigmoid → 0.889

0.8365 is the Helstrom accuracy on the test set

```
D1C = ToDensity[Normalize[Append[#, 1]]&/@X1];
D2C = ToDensity[Normalize[Append[#, 1]]&/@X2];
{p1C, p2C} = BinaryClassifier[D1C, D2C];
y1 = HelstromClassify[{p1C, p2C}, D1C];
y2 = HelstromClassify[{p1C, p2C}, D2C];
accuracy[y1, y2]
Show[Plot3D[{Tr[p1C.ToDensity[{Normalize[{x1, x2, 1}]}]][[1]]],
Tr[p2C.ToDensity[{Normalize[{x1, x2, 1}]}]][[1]]]}, {x1, −10, 10}, {x2, −10, 10},
PlotStyle → {Green, Red}, ViewPoint → {0, 0, ∞}, Lighting → {{"Ambient", White}}, Mesh → False],
Graphics3D[{RGBColor[0, 1, 0, 0.5], PointSize[0.013], Point[Append[#, 1]&/@X1]}],
Graphics3D[{RGBColor[1, 0, 0, 0.5], PointSize[0.013], Point[Append[#, 1]&/@X2]}]]
y = Join[y1, y2];
HelstromX1 = Pick[X, y, 1];
HelstromX2 = Pick[X, y, 2];
svm = Classify[X → y, Method → {"SupportVectorMachine", "KernelType" → "Linear"}];
Graphics[{Point[X], Blue, PointSize[Large], Point[getvectors[svm, X]], Opacity[0.5], Gray, getplane[svm, X]}]
AssociationMap[ClassifierMeasurements[Classify[X → y, Method → {"SupportVectorMachine", "KernelType" → #}],
1 → HelstromX1, 2 → HelstromX2, "Accuracy"]&, {"Linear", "RadialBasisFunction", "Polynomial", "Sigmoid"}]
```

0.503 is the Helstrom accuracy on the training set

$\text{Linear} \to 0.9925, \text{RadialBasisFunction} \to 0.9905, \text{Polynomial} \to 0.9955, \text{Sigmoid} \to 0.8655$

Heltrom classifier gives the best average success probability $(1/2+1/2(1/2\,|\rho_1 - \rho_2|_1)$, but it does not work with the same centroid (such as $1/2\,I$).

```
X1 = Table[{Tan[θ], 0}, {θ, 0, 2π, 2π/99}];
X2 = −RotateLeft[#, 1]&/@X1;
ListPlot[{X1, X2}, PlotStyle → Directive[PointSize[0.013], Opacity[0.7]], AspectRatio → 1, Frame → True, Axes → False]
D1C = ToDensity[Normalize[Append[#, 1]]&/@X1];
D2C = ToDensity[Normalize[Append[#, 1]]&/@X2];
```
$$\text{MatrixForm}[\#]\&/@N\left[\left\{\frac{\text{Length}[\text{D1C}]}{\text{Length}[\text{D1C}]+\text{Length}[\text{D2C}]}\text{Total}[\text{D1C}] − \frac{\text{Length}[\text{D2C}]}{\text{Length}[\text{D1C}]+\text{Length}[\text{D2C}]}\text{Total}[\text{D2C}], \frac{\text{Total}[\text{D1C}]}{\text{Length}[\text{D1C}]}, \frac{\text{Total}[\text{D2C}]}{\text{Length}[\text{D2C}]}\right\}\right]$$
```
AssociationMap[ClassifierMeasurements[Classify[1 → X1, 2 → X2, Method → #], 1 → X1, 2 → X2, "Accuracy"]&,
{"RandomForest", "NaiveBayes", "SupportVectorMachine", "NearestNeighbors"}]
```

$$\begin{pmatrix} \begin{pmatrix} 0 & 0 \\ 0 & 0 \end{pmatrix} \\ \begin{pmatrix} 0.82014 & 0 \\ 0 & 0.17986 \end{pmatrix} \\ \begin{pmatrix} 0.82014 & 0 \\ 0 & 0.17986 \end{pmatrix} \end{pmatrix}$$

$\text{RandomForest} \to 0.975, \text{NaiveBayes} \to 0.955, \text{SupportVectorMachine} \to 0.83, \text{NearestNeighbors} \to 0.985$

$$X1 = N\left[\text{Flatten}\left[\text{Table}\left[\left\{\left\{\frac{\text{Cot}[\theta]+\text{Csc}[\theta]}{\sqrt{2}}, \frac{\text{Tan}\left[\frac{\theta}{2}\right]}{\sqrt{2}}\right\}, \left\{-\frac{\text{Cot}\left[\frac{\theta}{2}\right]}{\sqrt{2}}, \frac{(-1+\text{Cos}[\theta])\text{Csc}[\theta]}{\sqrt{2}}\right\}\right\}, \left\{\theta, \frac{\pi}{100}, \pi - \frac{\pi}{100}, \frac{2\pi}{100}\right\}\right], 1\right]\right];$$

$$X2 = N\left[\text{Flatten}\left[\text{Table}\left[\left\{\left\{\frac{\text{Tan}\left[\frac{\theta}{2}\right]}{\sqrt{2}}, -\frac{\text{Cot}\left[\frac{\theta}{2}\right]}{\sqrt{2}}\right\}, \left\{\frac{(-1+\text{Cos}[\theta])\text{Csc}[\theta]}{\sqrt{2}}, \frac{\text{Cot}[\theta]+\text{Csc}[\theta]}{\sqrt{2}}\right\}\right\}, \left\{\theta, \frac{\pi}{200}, \pi - \frac{\pi}{200}, \frac{2\pi}{100}\right\}\right], 1\right]\right];$$
```
ListPlot[{X1, X2}, PlotStyle → Directive[PointSize[0.013], Opacity[0.7]], AspectRatio → 1, Frame → True, Axes → False]
D1 = SVMEncoder[#, 1]&/@X1;
D2 = SVMEncoder[#, 1]&/@X2;
{p1, p2} = BinaryClassifier[D1, D2];
y1 = HelstromClassify[{p1, p2}, D1];
y2 = HelstromClassify[{p1, p2}, D2];
accuracy[y1, y2]
Show[Plot3D[{Tr[p1.SVMEncoder[{x1, x2}, 1]], Tr[p2.SVMEncoder[{x1, x2}, 1]]}, {x1, −10, 10}, {x2, −10, 10},
PlotStyle → {Green, Red}, ViewPoint → {0, 0, ∞}, Lighting → {{"Ambient", White}}, Mesh → False],
Graphics3D[{RGBColor[0, 1, 0, 0.5], PointSize[0.013], Point[Append[#, 1]&/@X1]}],
```

$\text{Graphics3D}[\{\text{RGBColor}[1,0,0,0.5],\text{PointSize}[0.013],\text{Point}[\text{Append}[\#,1]\&/@X2]\}]]$

$\text{MatrixForm}[\#]\&/@\text{Chop}\left[N\left[\left\{\frac{\text{Length}[D1]}{\text{Length}[D1]+\text{Length}[D2]}\text{Total}[D1]-\frac{\text{Length}[D2]}{\text{Length}[D1]+\text{Length}[D2]}\text{Total}[D2],\frac{\text{Total}[D1]}{\text{Length}[D1]},\frac{\text{Total}[D2]}{\text{Length}[D2]}\right\}\right]\right]$

$\text{AssociationMap}[\text{ClassifierMeasurements}[\text{Classify}[1\rightarrow X1,2\rightarrow X2,\text{Method}\rightarrow\#],1\rightarrow X1,2\rightarrow X2,\text{"Accuracy"}]\&,$

$\{\text{"RandomForest"},\text{"NaiveBayes"},\text{"SupportVectorMachine"},\text{"NearestNeighbors"}\}]$

0.73 is the Helstrom accuracy on the training set

$$\left(\begin{array}{c}\begin{pmatrix}0.250031 & 12.5 & 0\\ 12.5 & -0.250031 & 0\\ 0 & 0 & 0\end{pmatrix}\\ \begin{pmatrix}0.375 & 0.125 & 0\\ 0.125 & 0.375 & 0\\ 0 & 0 & 0.25\end{pmatrix}\\ \begin{pmatrix}0.369999 & -0.125 & 0\\ -0.125 & 0.380001 & 0\\ 0 & 0 & 0.25\end{pmatrix}\end{array}\right)$$

$\text{RandomForest}\rightarrow 0.98,\text{NaiveBayes}\rightarrow 0.545,\text{SupportVectorMachine}\rightarrow 0.45,\text{NearestNeighbors}\rightarrow 0.85$

**FeatureVector::usage = "FeatureVector[m,n,type],**

**where m is the dimension of the input vector, n is the number of copies of the density operator, type of the SVMEncoder**

**\n\t Returns the feature vector.";**

$\text{FeatureVector}[m\_,n\_,\text{type}\_]:=\text{Block}[\{\rho,\text{state},x\},$

$\rho=\text{SVMEncoder}[\text{Array}[x,m],\text{type}];$

$\text{state}=\text{Nest}[\text{TensorProductQD}[\rho,\#]\&,\rho,n-1];$

$\text{DeleteDuplicates}[\text{DeleteCases}[\text{BlochVector}[\text{state}],0]];$

$\text{FeatureVector}[2,1,2]$

$$\left(\begin{array}{c}\frac{8x[1]x[2]}{\left(1+x[1]^2+x[2]^2\right)^2}\\ \frac{4x[1]\left(-1+x[1]^2+x[2]^2\right)}{\left(1+x[1]^2+x[2]^2\right)^2}\\ \frac{4x[2]\left(-1+x[1]^2+x[2]^2\right)}{\left(1+x[1]^2+x[2]^2\right)^2}\\ \frac{4x[1]^2}{\left(1+x[1]^2+x[2]^2\right)^2}-\frac{4x[2]^2}{\left(1+x[1]^2+x[2]^2\right)^2}\\ \frac{4x[1]^2}{\sqrt{3}\left(1+x[1]^2+x[2]^2\right)^2}+\frac{4x[2]^2}{\sqrt{3}\left(1+x[1]^2+x[2]^2\right)^2}-\frac{2\left(-1+x[1]^2+x[2]^2\right)^2}{\sqrt{3}\left(1+x[1]^2+x[2]^2\right)^2}\end{array}\right)$$

$\text{State}[n\_,x\_]:=\text{Block}[\{\rho,\text{state}\},$

$\rho=\text{SVMEncoder}[x,1];$

$\text{state}=\text{Nest}[\text{TensorProductQD}[\rho,\#]\&,\rho,n-1]];$

$\text{HelstromAccuracy}[n\_,X1\_,X2\_]:=\text{Block}[\{D1,D2,p1,p2,y1,y2,\text{acc}\},$

$\text{acc}=\{\};\text{Do}[$

$D1=\text{State}[i,\#]\&/@X1;$

$D2=\text{State}[i,\#]\&/@X2;$

$\{p1,p2\}=\text{BinaryClassifier}[D1,D2];$

$y1=\text{HelstromClassify}[\{p1,p2\},D1];$

$y2=\text{HelstromClassify}[\{p1,p2\},D2];$

$\text{AppendTo}\left[\text{acc},\left\{\text{Length}[\text{FeatureVector}[2,i,1]],N\left[\frac{\text{Count}[y1,1]+\text{Count}[y2,2]}{\text{Length}[y1]+\text{Length}[y2]}\right]\right\}\right],\{i,n\}\right];$

$\text{Return}[\text{acc}]];$

$\text{acc}=\text{HelstromAccuracy}[6,X1,X2];$

$\text{ListLinePlot}[\text{acc},\text{AxesLabel}\rightarrow\{\text{"dimension"},\text{"accuracy"}\}]$

## Appendix B. Mathematica Notebook for the Quantum-Inspired NMC

Quantum-inspired NMC with the moons dataset defined in Appendix A

CentroidClassify[{mean1_, mean2_}, D_] :=

If[Tr[ConjugateTranspose[mean1 − #].(mean1 − #)] ≤ Tr[ConjugateTranspose[mean2 − #].(mean2 − #)], 1, 2] & /@ D;

mean1 = Mean[D1];

mean2 = Mean[D2];

y1 = CentroidClassify[{mean1, mean2}, D1];

y2 = CentroidClassify[{mean1, mean2}, D2];

accuracy[y1, y2]

B1 = BlochVector[#] & /@ D1;

B2 = BlochVector[#] & /@ D2;

mean1 = Mean[B1];

mean2 = Mean[B2];

CentroidClassify[{mean1_, mean2_}, B_, p_] := If[Norm[mean1 − #, p] ≤ Norm[mean2 − #, p], 1, 2] & /@ B;

y1 = CentroidClassify[{mean1, mean2}, B1, 2];

y2 = CentroidClassify[{mean1, mean2}, B2, 2];

accuracy[y1, y2]

0.865 is the accuracy of the NMC with trace distance between density operators

0.865 is the accuracy of the NMC with Euclidean distance between Bloch vectors

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
