# Peer review of "Support Vector Machines with Quantum State Discrimination"

_quantumrep, doi:10.3390/quantum3030032_

Round 1

Reviewer 1 Report

In this paper, the authors theoretically and numerically investigate the relations between the support vector machines (SVMs) and the quantum state discrimination proposed in Ref. [5]. Connecting quantum machine learning (ML) to the well-established classical ML schemes is critical to the understanding of both subjects. In particular, SVMs belong to an important class of so-called interpretable ML methods. Several works have revealed the interesting and general relations between the quantum ML schemes and SVMs. I believe this paper makes non-trivial contributions to such a topic, which is to reduce the computational cost by explicitly using the quantum-inspired kernel functions. Thus I think the manuscript should be published eventually. Before giving my recommendation, some issues of the current manuscript should be resolved.

  1. In general, the basics are introduced properly, which makes the manuscript quite friendly for reading. But I have some comments on the writing of other parts. In the introduction, the existing progresses made in connecting quantum ML and SVMs are not introduced at all. For quantum ML, the main steps are similar to the SVMs: the first step is to encode classical data to a higher-dimensional quantum space, and the second step is to calculate the distances in the quantum space with the chosen kernel. I would say that the connections between quantum ML and SVMs are natural and established. Meanwhile, the quantum versions of the SVMs are also worth mentioning in the introduction. The authors may refer to Phys. Rev. Lett. 122, 040504 (2019), PRB 101, 075135 (2020), arXiv:2101.11020, to name just a few.
  2. For the numerical results, I wonder very much why the authors chose to put all of them in the appendix as a mathematical notebook. I can only find some brief analyses on the numerical data in the conclusion section. I strongly suggest to move the numerical data to a new section in the main text, and expand the analyses.

  3. A technical question: comparing the original Helstrom classifier and the SVM version of the Helstrom classifier, are they mathematically equivalent (except for the computational complexity)?

Author Response

We thank the referees for their helpful suggestions to improve the paper.
1. We added some existing progresses made in connecting quantum ML and SVMs in the introduction.
2. We moved the numerical data in a new section and expanded the analyses.
3. In the examples SVM behaves like the Helstrom algorithm, but is not equivalent.

Reviewer 2 Report

The work of R. Leporini and D. Pastorello, entitled: "Support vector machines with quantum state discrimination" presents a comprehensive analysis of some quantum-enabled classification methods and their connection with support vectors machines (SVM). Although, in my considered opinion, it is well know that SVM algorithms can be implemented in quantum machines by mapping classical data into quantum states, I believe the work of the authors could be useful for the research community interested in quantum-enabled machine learning.

However, I cannot recommend publication of the manuscript in its present form. Below follows a list of the points that need to be addressed before considering publication:

1) The introduction is rather short. Considering the vast literature on quantum (and classical) machine learning and state discrimination, I believe the authors need to present a more thorough analysis of what has been done in terms of comparison between classical SVM and quantum SVM.

2) Related to my previous point. It would be worth citing actual experimental work dealing with classification/discrimination of quantum states, see for instance:

Phys. Rev. Lett. 120, 240501 (2018)

New J. Phys. 22, 045001 (2020)

Phys. Rev. Lett. 124, 160401 (2020)

Appl. Phys. Rev. 7, 021404 (2020)

Adv. Quantum Technol. 3, 2000067 (2020)

3) The main claim of the authors, stated in the abstract, is that they "show how to make the classification more efficient in terms of space and time;" however, they do not present any quantitative analysis of such claim. They use phrases like: "more efficient", "impressive results", "improving the quality" but no quantitative measure of that is presented.  It seems like this information is somehow hidden in the appendix. I urge the authors to include a table or a list of the methods with their corresponding efficiency and execution time. Information about the computer where these algorithms are tested could be useful.

4) Although it is the authors' decision, I honestly believe that including the code in the appendix is not necessary and deviates the attention of the reader. I would suggest to remove all that information and just include the link to a permanent repository where readers could access the code. Keep in mind that copy-pasting the code could lead to errors in the process, making the life of readers very hard!

Author Response

We thank the referees for their helpful suggestions to improve the paper.
1. We added some existing progresses made in connecting quantum ML and SVMs in the introduction.
2. We included the suggested references.
3. We moved the numerical data in a new section and expanded the analyses.
4. We also put the code in a repository: https://github.com/leporini/classification

Round 2

Reviewer 1 Report

In the revised version, the authors have taken care of most of the comments raised in the last report. I just have one more question concerning the data. From Eq. (19), it seems that the accuracy is sensitive to the dimension. I would suggest to add a figure or table to show the accuracy with different dimensions.

Author Response

Thanks for your suggestion.
We added the figure to show that the accuracy is dimension sensitive and we updated the Mathematica code.

Reviewer 2 Report

The authors have addressed all my previous points.

I would just like to remark that there is an error in the references that I suggested. In Ref. [20] Appl. Phys. Rev. stands for Applied Physics Reviews, not "Applications Phys. Rev."

Author Response

Thanks we have updated the journal name.